# Artificial intelligence in orthopaedics: A scoping review

**Simon J. Federer**¤*, **Gareth G. Jones**

MSk Lab, Sir Michael Uren Hub, Imperial College London, London, United Kingdom

☯ These authors contributed equally to this work.
¤ Current address: Department of Trauma & Orthopaedics, St Richard's Hospital, Chichester, United Kingdom
* simon.federer@nhs.net

**Data Availability Statement:** The dataset may be found at this URL: https://data.mendeley.com/datasets/xvkr6t263v/1.

**Funding:** The author(s) received no specific funding for this work.

## Abstract

There is a growing interest in the application of artificial intelligence (AI) to orthopaedic surgery. This review aims to identify and characterise research in this field, in order to understand the extent, range and nature of this work, and act as springboard to stimulate future studies. A scoping review, a form of structured evidence synthesis, was conducted to summarise the use of AI in orthopaedics. A literature search (1946–2019) identified 222 studies eligible for inclusion. These studies were predominantly small and retrospective. There has been significant growth in the number of papers published in the last three years, mainly from the USA (37%). The majority of research used AI for image interpretation (45%) or as a clinical decision tool (25%). Spine (43%), knee (23%) and hip (14%) were the regions of the body most commonly studied. The application of artificial intelligence to orthopaedics is growing. However, the scope of its use so far remains limited, both in terms of its possible clinical applications, and the sub-specialty areas of the body which have been studied. A standardized method of reporting AI studies would allow direct assessment and comparison. Prospective studies are required to validate AI tools for clinical use.

## Introduction

Interest in the application of artificial intelligence (AI) in healthcare has surged in recent years [1]. Computer systems are increasingly able to perform tasks that normally require human intelligence, facilitated by improvements in data storage and computer processing. Despite the interest, incorporation of AI into clinical practice is in its infancy [2]. AI tools are currently in use, for example; in segmentation of three-dimensional optical coherence tomography scans to aid referrals in ophthalmology [3], and detection of atrial fibrillation by a smartphone algorithm and a single lead electrocardiography device in primary care [4]. The increase in digital medical imaging and information collected in databases and orthopaedic registries, provide large datasets ideal for the development of AI algorithms. These have the potential to improve patient care at a number of levels including; diagnosis, management, research and systems analysis [5].

**Competing interests:** The authors have declared that no competing interests exist.

The volume and variety of data collected from individuals has facilitated the advancement of AI across multiple industries. Concerns regarding how personal data is stored and utilised prompted legislation to protect this information. The General Data Protection Regulation (GDPR) was introduced in the European Union (EU) in 2018, and some medical registries have struggled to gather data in the same volume since. However, registries where patient consent has been a priority, such as the National Joint Registry (NJR) in the UK, have not seen a sharp decrease. The NJR holds information on over 3 million arthroplasty procedures since 2003 [6]. Orthopaedic registries are some of the largest in healthcare and are primed for the application of AI.

Artificial intelligence remains a relatively new field for most orthopaedic surgeons, and understanding the extent, range and nature of work conducted so far is useful as a springboard to identify potential new applications and areas for research. With this goal in mind, we conducted a scoping review, which is a form of structured evidence synthesis suited to this task. The aims were to: 1) identify the number of research studies using AI in orthopaedics and 2) summarize how and where these studies have applied AI to the field of orthopaedics.

## Methods

A scoping review was chosen due to the breadth of the research topic and the expected variation in study design, and was conducted using the Arksey and O'Malley framework [7]. The PRISMA-ScR checklist was utilised to ensure completeness (S1 Table) [8].

### Literature search and eligible studies

A literature search of studies in English was conducted (1946–2019) using Ovid (Embase & Medline) and Scopus. The search timeframe was chosen to ensure early studies were not missed. The literature search was performed on 30/8/19. The search strategy is shown in Fig 1. The search terms used are shown in S2 and S3 Tables.

The review focused on summarising the use of AI in applications relevant to clinical practice rather than related basic science. Hence, the following inclusion criteria were used: (a) studies which directly applied artificial intelligence to orthopaedic clinical practice or b) the outcomes of the study had the potential to be directly applied to orthopaedic clinical practice.

Abstracts, conference proceedings, articles not in English and review, commentary or editorial articles were not eligible for inclusion. Articles relating to the following were also excluded: cancer/oncology, biomechanics, gait analysis without clinical application, image segmentation alone without a direct clinical application, basic science, neuromuscular disorders, rehabilitation, prosthetics, natural language processing of radiology reports and wearable sensors. These articles were excluded to ensure the review maintained a clinical focus and was applicable to a general orthopaedic audience.

The literature search was performed by one investigator (SF). Abstract screening and full text reviews were performed independently by two investigators (SF and GJ). There was full agreement on the studies selected for inclusion. References from the literature search were imported into Mendeley (v1.19.6, Elsevier, Amsterdam, Netherlands) where duplicates were removed. Covidence systematic review software (Veritas Health Innovation Ltd, Melbourne, Australia. Available at www.covidence.org) was used to synthesize and extract eligible studies.

### Data extraction and collation

Data was extracted from eligible studies into an evidence table to summarize the following: year of publication, country, area of body, procedure, health condition, orthopaedic care function, study design and number of patients. A formal quality appraisal of eligible studies was not performed as this is beyond the remit of a scoping review. The data collected in the

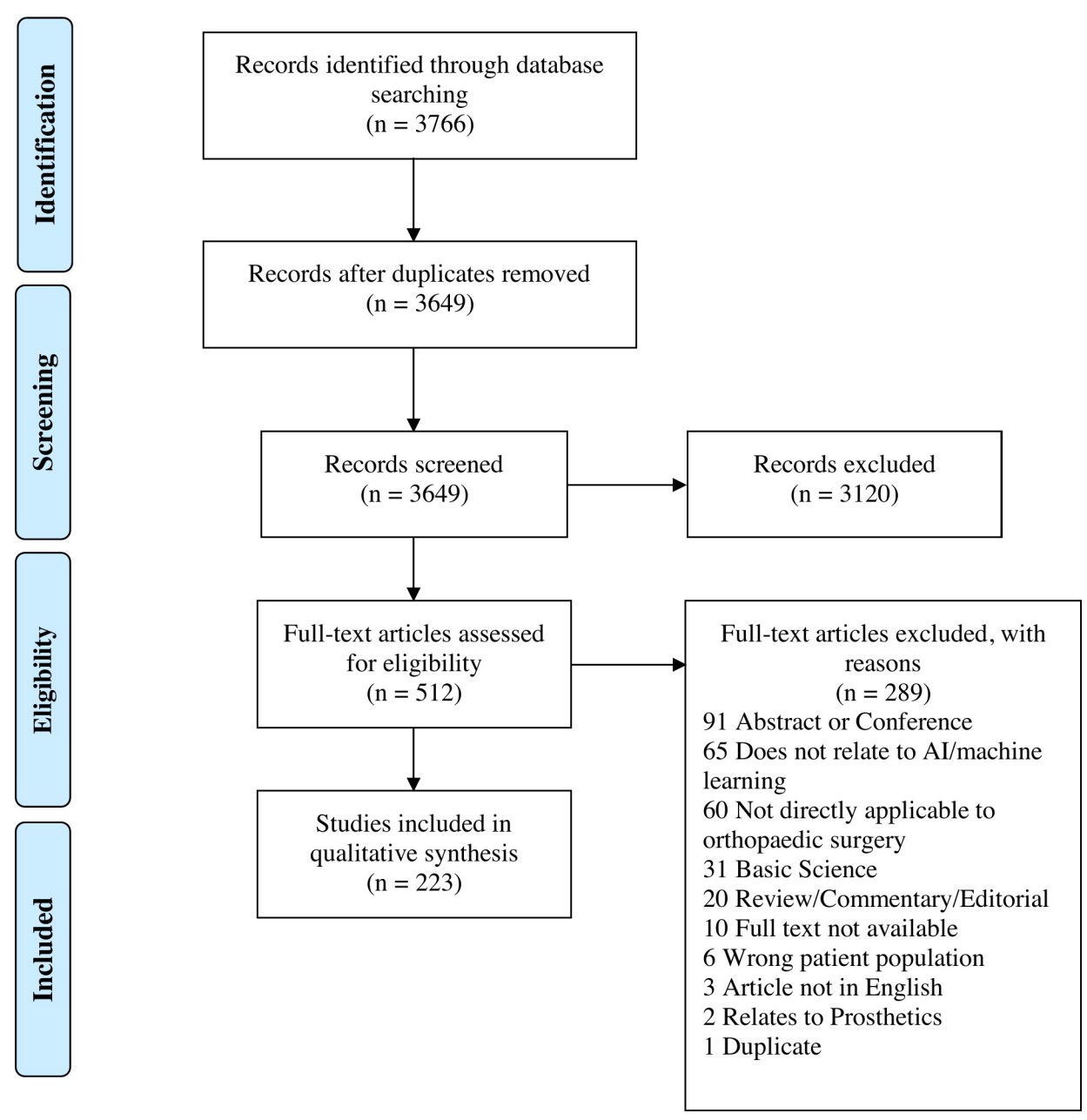

**Fig 1. Literature search and study identification strategy.** PRISMA flow diagram showing the search strategy and number of included and excluded studies.

evidence table was used to define the main themes of research and the summarised data represented below.

## Results

### Searches

After removal of duplicates, the search retrieved 3649 documents for title and abstract screening. Of those, 512 met the eligibility criteria for full text screening and 222 met the final

inclusion criteria. A reference list of included studies can be found in S4 Table. The study with the earliest publication date, 1989, used a machine learning method (inductive learning) to predict operative findings of disc prolapse or nerve entrapment [9]. 139 studies used one AI technique and 83 used more than one. Machine learning techniques were used 236 times and deep learning techniques 162 times. The most used machine learning techniques were Support Vector Machines, 55 times, and Random Forests, 38 times. Of the studies that used deep learning techniques, 26 implemented convolution layers in their neural networks. Characteristics of all the studies are summarised below in categories of data extraction.

## Imaging

101 studies used AI to interpret an imaging modality to establish a diagnosis. A number of early papers assessed and quantified the curvature of the spine in scoliosis [10–12], and developed algorithms capable of calculating the Cobb angle using surface topography before using radiographs and three-dimensional imaging. Subsequently, AI was applied to the detection of other spinal pathologies e.g disc herniation or vertebral fractures [13–16]. More recently the scope of AI to aid diagnostic imaging has expanded outside of the spine, with uses ranging from the identification of hip fractures to soft tissue meniscal tears in the knee [17–19]. There has also been a shift to algorithms providing a more nuanced grading of disease, rather than binary outputs [20].

## Orthopaedic care function

106 studies used AI to aid diagnostic decision support and 95 studies used AI to predict an aspect of a patient's care. The first paper to use AI in orthopaedics predicted operative findings during low back surgery [9]. The data comprised of preoperative clinical features and was analysed using an inductive learning method. More recently, research has focused on algorithms predicting patient outcomes post-surgery, utilizing the large orthopaedic data sets collected at local and national level. In particular, two centres in the USA have developed algorithms using local hospital data across different patient groups and procedures [21–25].

## Area of body

96 studies focused on the spine, 51 on the knee, 31 on the hip and 24 involved multiple areas. Other areas had 5 publications or fewer (Fig 2).

## Health condition

68 publications related to spinal pathologies, 64 to trauma and 62 to arthritis. Other conditions were reported in 5 studies or fewer.

## Procedure

141 publications did not relate to a specific orthopaedic procedure. 34 related to arthroplasty and 26 to spinal procedures. Other procedures were reported in 5 publications or fewer.

## Size of dataset used

There was a large range in the size of dataset used in the studies. The largest dataset used 1106234 patients [26], the smallest only 4 [27]. The median number of patients used was 250. 68 studies had a dataset of fewer than 100 patients. Arthroplasty registries were the sources of some of the larger datasets with information from over 1 million patients being used to build AI models [23, 26, 28–31].

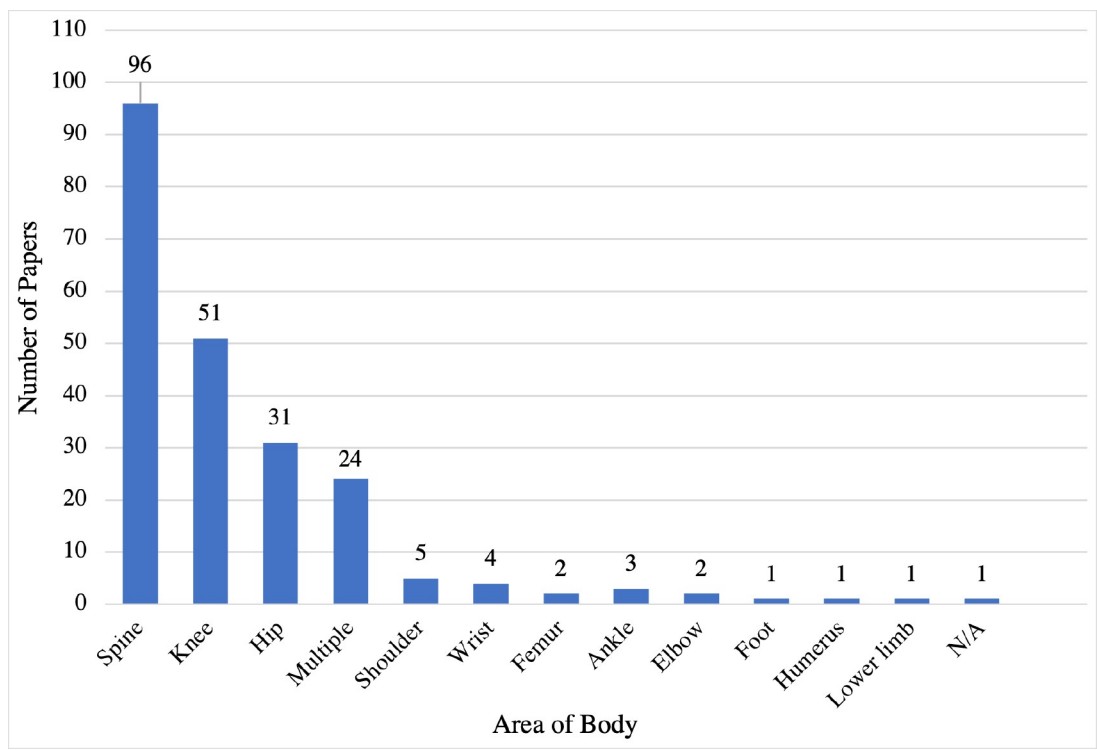

**Fig 2. Publication count by orthopaedic area of interest.** A graph showing the number of papers published with regards to the area of the body.

### Year of publication

The number of studies has increased in the last half a decade, with 14 publications in 2016 and 70 in 2019 (Fig 3). Between 1989 and 2010 the maximum number of publications per year was 6.

### Geographical location

83 studies (37%) were published from the USA, 24 from Canada, 23 from China, 11 from South Korea and 10 from India. Other countries had fewer than 10 published studies (Fig 4). Several papers from the USA emanate from the same institution, who have applied similar AI models to a range of applications [24, 32, 33].

### Discussion

We have reviewed and summarised the characteristics of 222 publications that included AI and orthopaedics. This scoping review was conducted to establish where and how AI has been used in orthopaedics. We have described the overarching features of these publications to highlight where the research has been focused and guide future avenues of research. The predominant findings were 1) Nearly half of the publications related to imaging interpretation to establish a diagnosis; 2) The spine was the most studied musculoskeletal region; and 3) Predicting patient outcomes is an emerging area of interest. Overall, research in AI and orthopaedics is at an early stage when compared to radiology [34], for example, but entering a phase of significant growth.

AI was used in 101 publications (45%) to interpret an imaging modality to establish a diagnosis. This focus can be explained by the large volume of organized data acquired during imaging and the relative ease with which AI models can be built to interpret this data. Radiology,

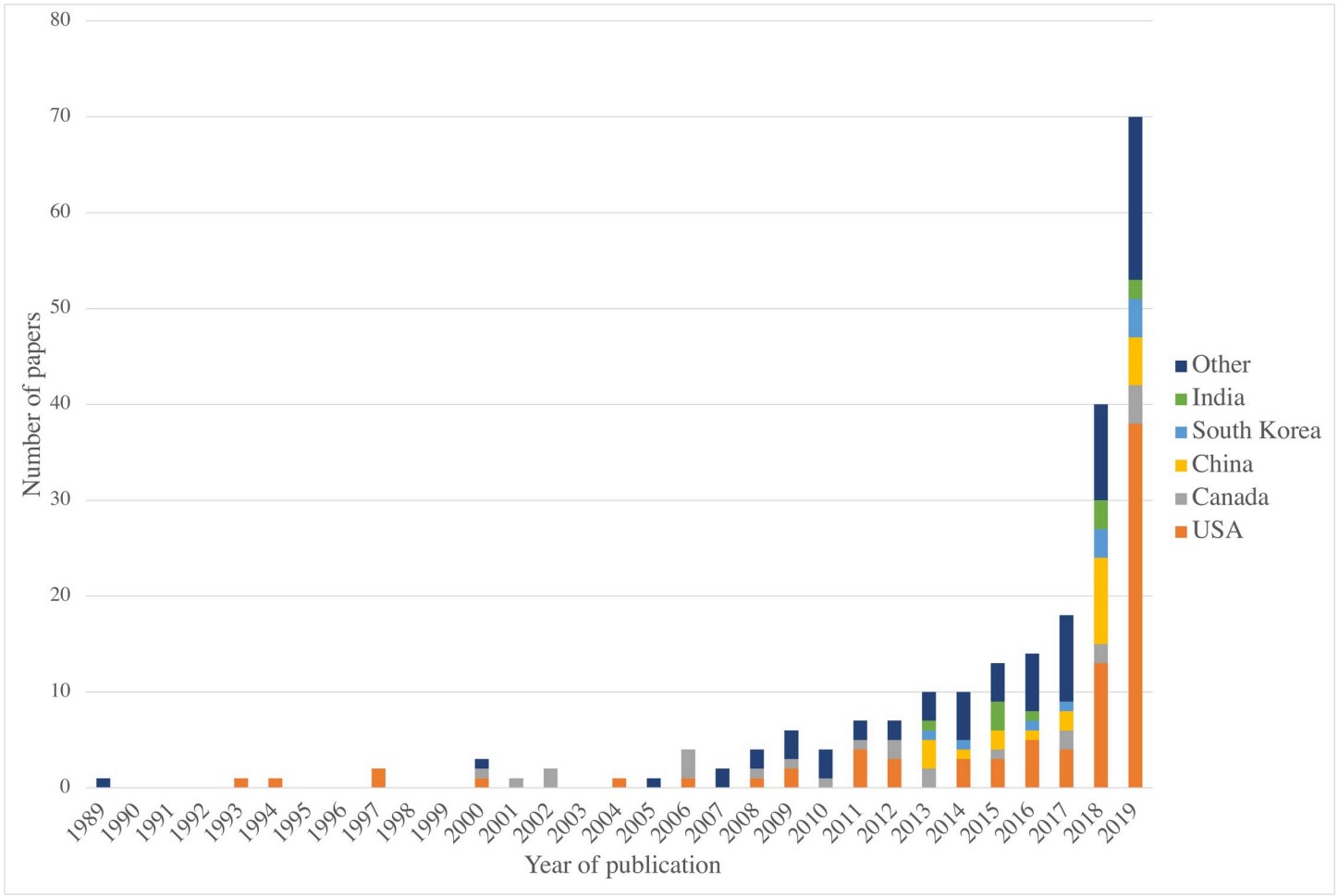

**Fig 3. Number of papers by year of publication and by country of origin.** A graph showing the number of papers published per year and by the country of origin. The five countries with the most publications are listed. Countries with fewer than 10 publications have been grouped into 'Other'.

accordingly, has seen one of the biggest increases in the use of AI to interpret scans [34]. The overlap between radiology and orthopaedics, for example, in fracture detection [13] or Cobb angle measurement from radiographs [35] could also explain the predominance of imaging related studies.

The initial search identified many publications relating to image segmentation, whereby an algorithm is used to automatically segment a specific structure(s), such as an intervertebral disc, from an imaging modality [36]. Papers that described segmentation of normal scans or were unable to detect pathology were not felt to be of direct clinical relevance and hence were excluded. Segmentation is, however, an important step in the process of establishing a diagnosis from imaging and it is relevant to mention the volume of research to date in this area. The use of real-time image segmentation with augmented reality is now being used as a navigation tool in spinal surgery [37], and this technique could be applied elsewhere in orthopaedics.

The spine, hip and knee were the regions most studied. The joint management of spinal pathology with neurosurgery could explain the greater proportion of papers on the spine. Large arthroplasty registries could suggest why hip and knee have seen more interest than the sub-specialty areas of foot & ankle and hand. More research should be focused on sub-specialty areas other than spine, hip and knee.

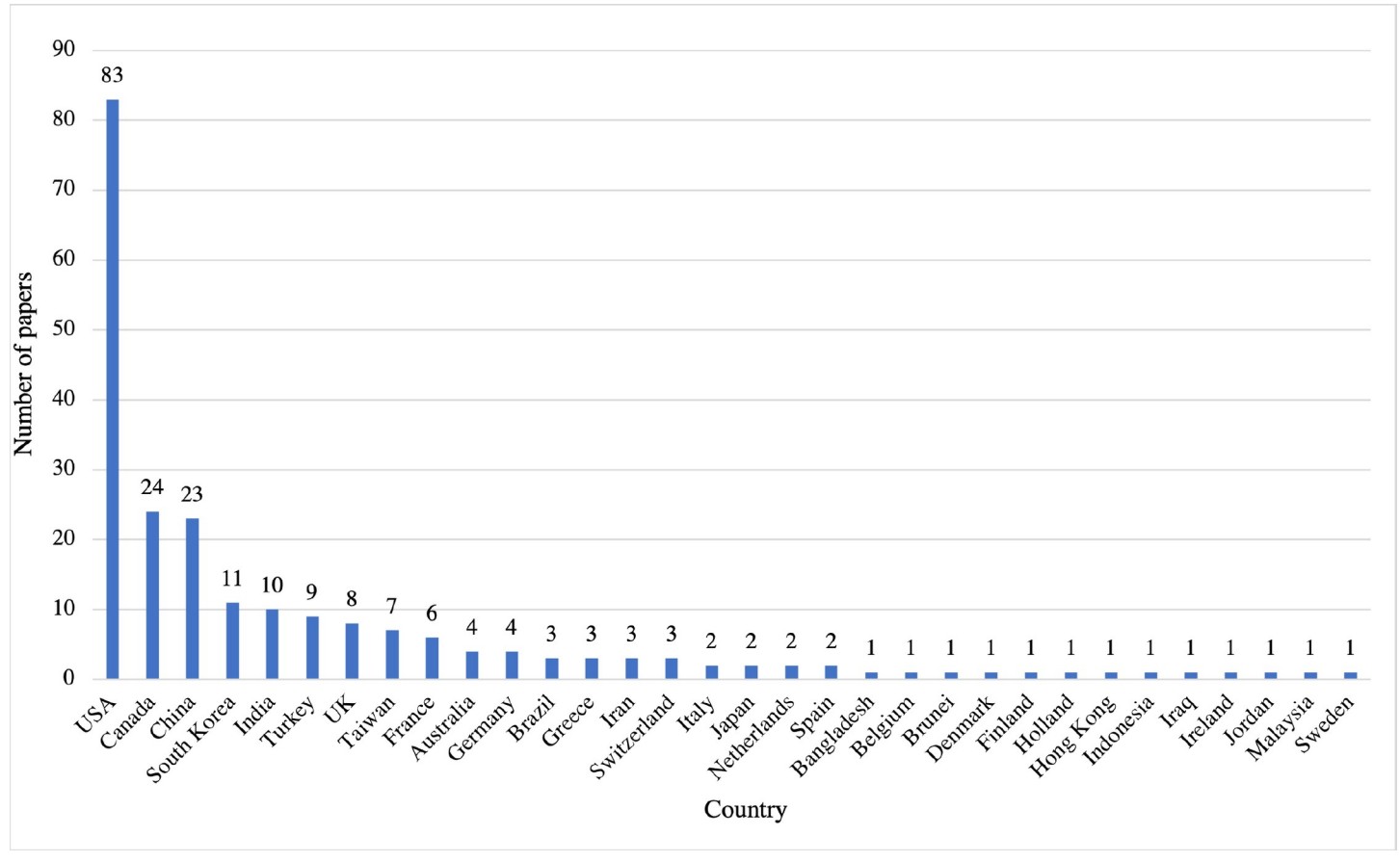

**Fig 4. Number of papers by country of origin.** A graph showing the number of papers published by the country of origin of the first author.

A significant volume of research found through the literature search related to translational engineering. A number of studies were published in engineering journals and so may not have reached readers from a clinical background [38–40]. Comparatively few papers from rheumatology were found in this study [41, 42]. This may be due to the inclusion criteria used and the health conditions of interest. The interplay between different specialties and industries presents an opportunity to promote interdisciplinary research. Specialists in data science are needed to progress AI in healthcare, and joint projects between specialties will make future research more efficient.

AI works best with high quality, large datasets. It was noted that the size of dataset in the published literature was highly variable. Sixty-eight (31%) of the studies had fewer than 100 patients. Whilst there is no set minimum dataset size for AI algorithms, the reliability of studies performed using small numbers may be questioned. Registries provided the largest sources of data in publications identified in this study [23, 26, 28–31]. They will continue to be a valuable resource for further studies predicting personalised patient outcomes. Albeit, there is concern that population-based data may be unable to solve clinical problems at a patient level [43]. Data sharing is needed for ongoing training and improvement of AI algorithms [2]. Legislation, such as GDPR, ensures that consent for data sharing is obtained and appropriate security measures are in place for the storage of data. Data privacy and protection is of utmost importance going forward.

There is scope for AI tools to assist in decision making regarding the management of patients. AI models that have been developed to retrospectively look at registry data could be used to design prospective studies. A decision-making aide would be a useful adjunct, for example, in understanding which patients will have favourable outcomes after arthroplasty. Predictive models will also provide insights into cost savings and efficiencies that will be of interest to healthcare providers.

AI is a rapidly advancing discipline with new algorithmic models constantly in development, often described using new and different terminology. Machine learning, deep learning and neural networks are some of the terms encountered in the literature that come under the umbrella term of AI. This variation in terminology has led to differences in how the papers are keyworded and recorded in databases. A PubMed (PubMed.gov, National Center for Biotechnology Information, Bethesda, MD, USA) search of "Artificial Intelligence Orthopaedics" in August 2019 yielded a mere 120 results. It was clear that many appropriate papers were missed and led to refinement of the search strategy for this study. A standardised method of reporting AI studies is currently lacking and would allow direct assessment and comparison of studies. Similarly, consistency in terminology and keywords would allow researchers to search for relevant papers more easily. "Artificial Intelligence" is, perhaps, too broad, and not clearly defined to be used as an umbrella term for keyword searches. We propose that the umbrella term "Machine learning" should be included on all papers for standardisation.

There was a geographical split in the location of papers published. As represented in Fig 3, most papers (n = 83) originated from the USA, followed by Canada (n = 24) and China (n = 23). These results may have been skewed by our inclusion only of papers written in English but highlights the dominance of institutions from the USA. Additionally, it is important to note that the search terms, whilst broader than a previous literature review [1] were not exhaustive, and despite our best efforts valid publications may have been missed. Some time has passed since the literature search was performed, and progress has been made in AI in orthopaedics and more widely in healthcare. Efforts to quantify the diagnostic accuracy of deep learning in medical imaging and guidelines for reporting such studies are two examples of how the field has progressed [44, 45].

## Conclusion

The use of AI in orthopaedics is increasing. Studies using large datasets exist and novel AI tools with the ability to have clinical impact are being developed. More research is needed before the potential of AI can translate to a significant change in the day-to-day clinical practice of orthopaedic surgeons.

## Supporting information

**S1 Table. PRISMA-ScR checklist.**
(DOCX)

**S2 Table. Database search terms for Ovid—Embase and Medline.**
(DOCX)

**S3 Table. Database search terms for Scopus.**
(DOCX)

**S4 Table. Reference list of papers included in study.**
(DOCX)

## Author Contributions

**Conceptualization:** Gareth G. Jones.

**Data curation:** Simon J. Federer.

**Formal analysis:** Simon J. Federer.

**Investigation:** Simon J. Federer.

**Methodology:** Simon J. Federer.

**Project administration:** Gareth G. Jones.

**Supervision:** Gareth G. Jones.

**Writing – original draft:** Simon J. Federer.

**Writing – review & editing:** Gareth G. Jones.

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
