## [Decision Letter · Decision Letter 0]

8 Jul 2021

PONE-D-21-16475

Artificial intelligence in orthopaedics: a scoping review

PLOS ONE

Dear Dr. Federer,

Thank you for submitting your manuscript to PLOS ONE. After careful consideration, we feel that it has merit but does not fully meet PLOS ONE’s publication criteria as it currently stands. Therefore, we invite you to submit a revised version of the manuscript that addresses the points raised during the review process.

We look forward to receiving your revised manuscript.

Kind regards,

Georg Osterhoff, M.D.

Academic Editor

PLOS ONE

Journal Requirements:

Reviewers' comments:

Reviewer's Responses to Questions

**Comments to the Author**

1. Is the manuscript technically sound, and do the data support the conclusions?

Reviewer #1: Yes

Reviewer #2: Partly

2. Has the statistical analysis been performed appropriately and rigorously? 

Reviewer #1: Yes

Reviewer #2: N/A

3. Have the authors made all data underlying the findings in their manuscript fully available?

Reviewer #1: Yes

Reviewer #2: Yes

4. Is the manuscript presented in an intelligible fashion and written in standard English?

Reviewer #1: Yes

Reviewer #2: Yes

5. Review Comments to the Author

Reviewer #1: Introduction: There should be a concrete example of implementation of AI based decision support or image classification/segmentation to exemplary show the scope of new developments in this field. Additionally, the ethical and data privacy aspect of saving and processing patient data in big quantities should be addressed.

Material and methods:

It is unfortunate that the review stops at 2019, since the quantity of new papers covering this area of interest hast evolved since then. This should be discussed later.

The exclusion criteria seem very specific and the decision for exclusion should be explained concretely. I.e. exclusion rehabilitation and prosthetics can be applied in clinical application.

Results:

Short and focused on the few results given. It would be interesting to correlate the year of publication and geographical location of the papers, maybe a graph could be added to emphasize this.

Discussion:

A clear differentiation between classic algorithms and machine learning or artificial intelligence based self-learning systems needs to be implemented, since this is topic to controversial discussion, since we lack clear definitions to the term “artificial intelligence”.

Overlapping research fields: In conclusion this presents us with the opportunity to promote interdisciplinary research, an opportunity which thus far is underdeveloped, especially because specialist knowledge from data science is needed to progress in the field of machine learning and AI in the general field of medicine.

Conclusion:

Ethical considerations and data privacy, especially with “big data”, as needed for development of AI and ML (machine learning) need to be discussed. Line 207: “AI in clinical practice is embryoic” -> well put, but the research in this paper is not able to support this thesis well enough and it seems like an overdramatization. Other wording should be considered.

The conclusion given make general suggestions, that whilst very interessting, cannot be supported by a scoping review, with the few data end points extracted from the 222 papers included. More careful evaluation of results and more focused discussion and conclusion should be considered

Reviewer #2: Dear authors, thank you very much for this very interesting, timely manuscript. The present manuscript is well-written and aims to give an overview on this very relevant topic. Unfortunately, it is hard for me to identify the general concept throughout the paper. The structure should be revised and the sections “results”, “discussion” as well as “conclusion” should be extensively revised, focusing on the actual meaning of these sections. Furthermore, referencing must be optimized in order to support your statements. Thus, I am afraid that this paper should not be published in its present state and needs extensive revision. Below you will find some specific comments.

Specific comments:

Introduction:

Since the European General Data Protection Regulation, medical registries are struggling to gather data and therefore, there is actually a decrease of registry data in most of the European countries. Please take that into account and adjust to it.

Literature search

Was there full agreement on the studies selected between both authors? Please add whether there was full agreement or if there was agreement negotiated.

In table 2 you mention search-terms for MEDLINE but did not report literature search in medline for this section. Please adjust.

Please describe whether you have only searched for articles in English or any other language.

Discussion

Generally, I would prefer better structuring of the discussion. I would not put sub-headings and the discussion should be related to the results presented. Unfortunately, in my opinion, a clear concept of this paper is lacking.

You are mentioning that research of AI in orthopaedics is at an early stage. How can you support this? Is there more research in other medical fields?

In my opinion, parts of the discussion should be rather pointed out in the results section (e.g. line 141-149). Please adjust this to the whole manuscript (strictly describing results versus discussion of these findings).

Overlapping research fields: this sub-section seems a bit out of place. Unfortunately, I cannot identify the central idea behind this.

Please discuss the arising data protection problems in regards of registry data.

Please add references to support your statements (e.g. line 159-161).

Conclusion

In my opinion, your conclusion is more a discussion than a straight conclusion. Please adjust this and identify actual discussion of findings, compared to a straight conclusion based on your manuscript. Furthermore, I think you should point out the relevance of your findings more extensively.

6. PLOS authors have the option to publish the peer review history of their article (what does this mean?). If published, this will include your full peer review and any attached files.

Reviewer #1: **Yes: **Dr. med. David Baur

Reviewer #2: No

---

## [Author Response · Author response to Decision Letter 0]

17 Aug 2021

Dear Editor and Reviewers,

Thank you for your time reviewing and commenting on our submission. Please see the table below detailing where specific comments have been addressed in the revised manuscript.

Introduction

• There should be a concrete example of implementation of AI based decision support or image classification/segmentation to exemplary show the scope of new developments in this field. 

First paragraph of introduction

Line 42

• Additionally, the ethical and data privacy aspect of saving and processing patient data in big quantities should be addressed. 2nd paragraph of introduction

Line 52

• Since the European General Data Protection Regulation, medical registries are struggling to gather data and therefore, there is actually a decrease of registry data in most of the European countries. Please take that into account and adjust to it. 2nd paragraph of introduction

Line 52

Methods

• It is unfortunate that the review stops at 2019, since the quantity of new papers covering this area of interest hast evolved since then. This should be discussed later. 

Mentioned in discussion. Line 357

• The exclusion criteria seem very specific and the decision for exclusion should be explained concretely. I.e. exclusion rehabilitation and prosthetics can be applied in clinical application. Line 95

• Was there full agreement on the studies selected between both authors? Please add whether there was full agreement or if there was agreement negotiated Line 100

• In table 2 you mention search-terms for MEDLINE but did not report literature search in medline for this section. Please adjust. Line 79

• Please describe whether you have only searched for articles in English or any other language. Line 78 and 90

Results

• Short and focused on the few results given. It would be interesting to correlate the year of publication and geographical location of the papers, maybe a graph could be added to emphasize this. 

Stacked bar chart as updated figure 3

Discussion

• A clear differentiation between classic algorithms and machine learning or artificial intelligence based self-learning systems needs to be implemented, since this is topic to controversial discussion, since we lack clear definitions to the term “artificial intelligence”. 

First paragraph of results. Line 120

• Overlapping research fields: In conclusion this presents us with the opportunity to promote interdisciplinary research, an opportunity which thus far is underdeveloped, especially because specialist knowledge from data science is needed to progress in the field of machine learning and AI in the general field of medicine. Added to discussion. Line 236.

• Generally, I would prefer better structuring of the discussion. I would not put sub-headings and the discussion should be related to the results presented. Unfortunately, in my opinion, a clear concept of this paper is lacking. Acknowledged. Headings removed. Restructured.

• You are mentioning that research of AI in orthopaedics is at an early stage. How can you support this? Is there more research in other medical fields? Line 186.

• In my opinion, parts of the discussion should be rather pointed out in the results section (e.g. line 141-149). Please adjust this to the whole manuscript (strictly describing results versus discussion of these findings). Results & discussion restructured

• Overlapping research fields: this sub-section seems a bit out of place. Unfortunately, I cannot identify the central idea behind this. Acknowledged. Results & discussion restructured 

• Please discuss the arising data protection problems in regards of registry data.

 Discussed in introduction and again in line 319-322.

• Please add references to support your statements (e.g. line 159-161) References added. Now lines 192-195.

Conclusion

• Ethical considerations and data privacy, especially with “big data”, as needed for development of AI and ML (machine learning) need to be discussed. Line 207: “AI in clinical practice is embryoic” -> well put, but the research in this paper is not able to support this thesis well enough and it seems like an overdramatization. Other wording should be considered. Ethical considerations added. Lines 319-322. Conclusion amended.

• The conclusion given make general suggestions, that whilst very interessting, cannot be supported by a scoping review, with the few data end points extracted from the 222 papers included. More careful evaluation of results and more focused discussion and conclusion should be considered Conclusion amended to be more focused.

• In my opinion, your conclusion is more a discussion than a straight conclusion. Please adjust this and identify actual discussion of findings, compared to a straight conclusion based on your manuscript. Furthermore, I think you should point out the relevance of your findings more extensively. Conclusion amended.

We hope you find our revised manuscript suitable for publication and look forward to hearing from you in due course.

---

## [Decision Letter · Decision Letter 1]

1 Oct 2021

PONE-D-21-16475R1Artificial intelligence in orthopaedics: a scoping reviewPLOS ONE

Dear Dr. Federer,

Thank you for submitting your manuscript to PLOS ONE. After careful consideration, we feel that it has merit but does not fully meet PLOS ONE’s publication criteria as it currently stands. Therefore, we invite you to submit a revised version of the manuscript that addresses the points raised during the review process.

We look forward to receiving your revised manuscript.

Kind regards,

Georg Osterhoff, M.D.

Academic Editor

PLOS ONE

Journal Requirements:

Reviewers' comments:

Reviewer's Responses to Questions

**Comments to the Author**

1. If the authors have adequately addressed your comments raised in a previous round of review and you feel that this manuscript is now acceptable for publication, you may indicate that here to bypass the “Comments to the Author” section, enter your conflict of interest statement in the “Confidential to Editor” section, and submit your "Accept" recommendation.

Reviewer #1: All comments have been addressed

Reviewer #2: All comments have been addressed

2. Is the manuscript technically sound, and do the data support the conclusions?

Reviewer #1: Yes

Reviewer #2: Yes

3. Has the statistical analysis been performed appropriately and rigorously? 

Reviewer #1: Yes

Reviewer #2: Yes

4. Have the authors made all data underlying the findings in their manuscript fully available?

Reviewer #1: Yes

Reviewer #2: Yes

5. Is the manuscript presented in an intelligible fashion and written in standard English?

Reviewer #1: Yes

Reviewer #2: Yes

6. Review Comments to the Author

Reviewer #1: I thank the authors for adressing the comments made in the last review. With small adjustments the paper should be eligble to be published.

Introduction: Added examples and insight into the collection of data and the potential for those registries complements the overall picture well.

Literature search and eligible studies

No further comments.

Results:

Lines 118-122: Since the

The term classical algorithms should not be used in this way. It is to unspecific, rather use: machine learning tequniques, since VMs, random forests can be classified as such. Furthermore ANN artifical neural networs and CNN convolutional neural networks should not be separated in these two groups. ANN is a very unspecific terminology as well and can and does in many papers include CNNs. This should be adressed. i.e. Of the artificial networks XX implemented convolution layers... etc.

No further comments.

Discussion:

Line 286-287: Eventough I agree that terminology is very herterogenic when it comes to AI, the term Artificial intelligence is not clearly defined, therefore is not suted as an umbrella term for keyword searches. The problem I see is the broadness of this term, which would generate an unclear and broad term for researchers.

Conclusion:

No comments.

Reviewer #2: Dear authors, thank you very much for this revision. Thanking you for the amendments. Publication should be considered now.

7. PLOS authors have the option to publish the peer review history of their article (what does this mean?). If published, this will include your full peer review and any attached files.

Reviewer #1: No

Reviewer #2: No

---

## [Author Response · Author response to Decision Letter 1]

11 Oct 2021

Dear Editor and Reviewers,

Thank you for your time reviewing and commenting on our submission. The further comments and suggestions have been noted, and we have amended our submission accordingly.

Lines 118-122 have been updated with the appropriate terminology. Lines 245-248 have been amended to suggest “Machine learning” as a more suitable umbrella term.

We hope you find our revised manuscript suitable for publication and look forward to hearing from you in due course.

Yours sincerely, 

Simon Federer

---

## [Decision Letter · Decision Letter 2]

11 Nov 2021

Artificial intelligence in orthopaedics: a scoping review

PONE-D-21-16475R2

Dear Dr. Federer,

We’re pleased to inform you that your manuscript has been judged scientifically suitable for publication and will be formally accepted for publication once it meets all outstanding technical requirements.

Kind regards,

Thippa Reddy Gadekallu

Academic Editor

PLOS ONE

Additional Editor Comments (optional):

Reviewers' comments:

Reviewer's Responses to Questions

**Comments to the Author**

1. If the authors have adequately addressed your comments raised in a previous round of review and you feel that this manuscript is now acceptable for publication, you may indicate that here to bypass the “Comments to the Author” section, enter your conflict of interest statement in the “Confidential to Editor” section, and submit your "Accept" recommendation.

Reviewer #1: All comments have been addressed

2. Is the manuscript technically sound, and do the data support the conclusions?

Reviewer #1: Yes

3. Has the statistical analysis been performed appropriately and rigorously? 

Reviewer #1: Yes

4. Have the authors made all data underlying the findings in their manuscript fully available?

Reviewer #1: Yes

5. Is the manuscript presented in an intelligible fashion and written in standard English?

Reviewer #1: Yes

6. Review Comments to the Author

Reviewer #1: Abstract

No further comments.

Introduction

No further comments.

Methods

No further comments

Results

No further comments

Discussion

Comment was included. No further comments.

Conclusion

Short and on point. No further comments.

7. PLOS authors have the option to publish the peer review history of their article (what does this mean?). If published, this will include your full peer review and any attached files.

Reviewer #1: No

---

## [Editor Report · Acceptance letter]

15 Nov 2021

PONE-D-21-16475R2 

Artificial intelligence in orthopaedics: a scoping review 

Dear Dr. Federer:

I'm pleased to inform you that your manuscript has been deemed suitable for publication in PLOS ONE. Congratulations! Your manuscript is now with our production department. 

Kind regards, 

on behalf of

Dr. Thippa Reddy Gadekallu 

Academic Editor

PLOS ONE